# The Relationship between Suicidality and Socio-Demographic Variables, Physical Disorders, and Psychiatric Disorders: Results from the Singapore Mental Health Study 2016

**DOI:** 10.3390/ijerph18084365

**Published:** 2021-04-20

**Authors:** Kundadak Ganesh Kudva, Edimansyah Abdin, Janhavi Ajit Vaingankar, Boon Yiang Chua, Saleha Shafie, Swapna Kamal Verma, Daniel Shuen Sheng Fung, Derrick Heng Mok Kwee, Siow Ann Chong, Mythily Subramaniam

**Affiliations:** 1Institute of Mental Health, Singapore 539747, Singapore; edimansyah_abdin@imh.com.sg (E.A.); janhavi_vaingankar@imh.com.sg (J.A.V.); boon_yiang_chua@imh.com.sg (B.Y.C.); saleha_shafie@imh.com.sg (S.S.); Swapna_verma@imh.com.sg (S.K.V.); daniel_fung@imh.com.sg (D.S.S.F.); siow_ann_chong@imh.com.sg (S.A.C.); mythily@imh.com.sg (M.S.); 2Ministry of Health, Singapore 169854, Singapore; derrick_heng@moh.gov.sg

**Keywords:** demographic variables, physical disorders, psychiatric disorders, suicidality

## Abstract

Suicidality encompasses suicidal ideation, plans, and attempts. This paper aims to establish associations between suicidality and sociodemographic variables, physical disorders, and psychiatric disorders. The Singapore Mental Health Study 2016 was a population-level epidemiological survey, which determined the prevalence of physical disorders, psychiatric disorders, and suicidality. Questionnaires were used to determine socio-demographic information. A total of 6216 respondents were interviewed. Lifetime prevalence of suicidal ideation, planning, and attempts were 7.8%, 1.6%, and 1.6%, respectively. All components of suicidality were more likely in those with major depressive disorder, bipolar disorder, generalized anxiety disorder, alcohol use disorder, and chronic pain. Suicidal ideation and attempts were more likely in those with diabetes. Age above 65, being male, and a monthly household income of ≥ SGD 10,000 were associated with a lower likelihood of suicidal ideation. These findings indicate that there are high-risk groups for whom suicidality is a concern, and for whom interventions may be needed.

## 1. Introduction

Suicidality, which encompasses suicidal ideation, plans, and attempts, is a clinical concern and has an adverse impact on individuals, families, and society. Individuals with suicidal ideation [1], and those who have planned for a suicide attempt [2], are at greater risk of attempting suicide, with those who have demonstrated planning being at a higher risk than those with ideation [3]. A suicide attempt is the most potent risk factor for completed suicide [1]. Research has been conducted to identify the determinants of a suicide attempt and factors identified include depressive [4] and anxiety disorders [5], poor social adjustment [6], unemployment [7], and medical illness [8].

Singapore is a city-state in Southeast Asia and is a multi-ethnic nation. The Singapore population as of 2018 stands at 5.6 million, of whom 3.9 million are residents (Citizens and Permanent Residents) [9]. The suicide rate in Singapore between 2012 and 2016 stood at 9.14 deaths per 100,000 residents, with a significant rise in elderly suicides between 2011 and 2017 [10]. Despite the suicide rate being lower than that seen in other countries, there are ongoing multi-pronged national efforts to prevent suicides under the National Mental Health Blueprint, such as by encouraging early help seeking, supporting groups at-risk and ensuring community support [11]. The Singapore Mental Health Study, 2016 (hereafter referred to as SMHS 2016) was the second iteration of a national mental health study [12], the first of which was conducted in 2010 (hereafter referred to as SMHS 2010). The SMHS 2016 aimed to establish the prevalence of select psychiatric disorders and to track the state of mental health in the Singapore resident population. The current article describes the prevalence of suicidality (ideation, plan, and attempts) in Singapore’s population, and explores associations among this and demographic variables, physical disorders, and psychiatric disorders.

## 2. Methods

### 2.1. Sample

The methodology of SMHS 2016 was similar to SMHS 2010 and has been described previously [12]. SMHS 2016 was a cross-sectional nationwide epidemiological survey of Singapore residents aged 18 years and above. Participants were randomly drawn from a national population registry of all citizens and permanent residents. Disproportionate stratified sampling was used, with oversampling of individuals from minority (Malay and Indian) ethnicities and those above 65 years of age—this was done to achieve adequate sample sizes and to improve the reliability of estimates when conducting subgroup analyses. Following an invitation letter, selected participants were approached in their homes and written informed consent was obtained prior to conducting face-to-face interviews. All interviews were conducted in the language that participants were most comfortable with (i.e., English, Mandarin, or Malay). The study was approved by the institutional ethics committee (National Healthcare Group Domain Specific Review Board).

### 2.2. Measures

#### 2.2.1. WHO Composite International Diagnostic Interview (WHO-CIDI)

The WHO-CIDI was used as the main diagnostic instrument to establish the prevalence of psychiatric disorders. A fully structured, computer-assisted version was used. The WHO-CIDI establishes the 12-month and lifetime prevalence of common psychiatric disorders following the criteria stipulated in the Diagnostic and Statistical Manual of Mental Disorders (DSM-IV) [13]. The disorders included affective disorders (major depressive disorder and bipolar disorder), anxiety disorders (generalized anxiety disorder and obsessive-compulsive disorder), and alcohol use disorder (alcohol abuse and alcohol dependence). These disorders were chosen after consultation with stakeholders and policy makers (e.g., The Ministry of Health, Singapore, voluntary welfare organizations catering to people with mental illness, and clinicians).

Participants were also asked to complete, as part of the WHO-CIDI, questions pertaining to suicidality. The questions asked are listed in [App ijerph-18-04365]. The WHO-CIDI has been used to assess suicidality in developed and developing nations [14]. Questions were read out to participants who were unable to read, whilst those who could read were given cue cards to refer to.

#### 2.2.2. Socio-Demographic Questionnaire

Data on age, gender, ethnicity (Chinese, Malay, Indian, and others), marital status (single, married, divorced/separated, or widowed), highest attained educational level, household income and employment status (employed, unemployed, or economically inactive e.g., retired) was collected.

#### 2.2.3. Physical Disorders

A modified version of the CIDI chronic conditions checklist, where participants were asked if they have been diagnosed with a list of health conditions, was used to ascertain the presence of physical disorders. The conditions included in this checklist were hyperlipidemia, hypertension, diabetes mellitus, asthma, chronic pain (e.g., arthritis, back/spine problems, and migraine headaches), cardiovascular diseases (e.g., stroke and heart disease including heart attack/s, coronary heart disease, angina, and congestive heart failure), gastrointestinal ulcers (e.g., stomach ulcer, enteritis, or colitis), thyroid disease, and cancer).

### 2.3. Statistical Analysis

In order to account for the disproportionate sampling design and to ensure that survey findings were representative of the Singaporean adult population, all estimates were weighted to adjust for over-sampling, non-response, and post-stratified for age and ethnicity distributions between the survey sample and the Singapore resident population in 2014. Descriptive analyses were performed to describe the socio-demographic profile of the study population, and the prevalence of suicidal ideation, planning, and attempts. Socio-demographic correlates of suicidal ideation, planning, and attempts, were determined using multivariable logistic regression. All socio-demographic variables (i.e., age group, gender, ethnicity, education, employment, marital status, and income) were included as predictor variables. Associations between each component of suicidality and physical and psychiatric diseases were analyzed via multivariable logistic regression. In the multivariable regression models, each of suicidal ideation, planning, and attempts was individually treated as a dependent variable, and the occurrence of each physical or psychiatric disorder was treated as a main predictor variable while controlling for all socio-demographic variables. All analyses were conducted utilizing STATA version 15.1.

## 3. Results

### 3.1. Characteristics of Study Sample

SMHS 2016 included a total of 6126 respondents. The response rate amongst eligible adults was 69.5%. The mean age of the sample was 45.2 years, with an approximately equal proportions of both genders (male 50.1%, female 49.9%). Moreover, 3843 (62.7%) of the respondents were married, 2835 (46.3%) had attained primary and secondary school education, and 1455 (23.8%) attained university education. A total of 4055 (66.2%) reported being employed and 354 (5.8%) were unemployed. Table 1 presents the socio-demographic distribution of study participants.

### 3.2. Suicidal Ideation, Planning and Attempts, and Associated Variables

#### 3.2.1. Suicidal Ideation 

The lifetime prevalence of suicidal ideation was 7.8%, with a 12-month prevalence of 1.3%. Table 2A depicts the association between lifetime suicidal ideation and demographic variables, while Table 2B depicts the association between lifetime suicidal ideation and physical and psychiatric disorders. Suicidal ideation was less likely among those aged 50–64 (OR 0.4, *p*= 0.001), those above 65 years of age (OR 0.3, *p* = 0.002), male respondents (OR 0.7, *p* = 0.034), those of Malay ethnicity (OR 0.7, *p* = 0.028) and those with a monthly household income level of SGD 4000–5999 (OR 0.6, *p* = 0.042) and SGD 10,000 and above (OR 0.5, *p* = 0.037). Having never been married (OR 1.8, *p* = 0.007) and being divorced/separated (OR 3.6, *p* = 0.000) were associated with a higher prevalence of suicidal ideation. After controlling for the presence of psychiatric disorders, all the socio-demographic variables remained significant, except for age 50–64 (*p* = 0.055). Those with major depressive disorder (OR 4.8, *p* = 0.000), bipolar disorder (OR 11, *p* = 0.000), generalized anxiety disorder (OR 10.3, *p* = 0.000), obsessive-compulsive disorder (OR 5.4, *p* = 0.000) and alcohol use disorder (OR 3.1, *p* = 0.000) were more likely to have suicidal ideation. Those with diabetes mellitus (OR 1.9, *p* = 0.020), chronic pain (OR 2.2, *p* = 0.000), and cancer (OR 2.9, *p* = 0.020) were also more likely to report suicidal ideation.

#### 3.2.2. Suicide Planning

The lifetime prevalence of suicide planning was 1.6%, with a 12-month prevalence of 0.3%. Table 3A depicts the association between suicidal planning and demographic variables, while Table 3B depicts the association between suicidal planning and physical and psychiatric disorders. Compared to those who had completed university, individuals with a highest attained educational level of pre-university/junior college were less likely to have engaged in suicide planning (OR 0.2, *p* = 0.030), and this remained significant (*p* = 0.048) after controlling for the presence of psychiatric disorders. Those with major depressive disorder (OR 5.4, *p* = 0.000), bipolar disorder (OR 11, *p* = 0.000), generalized anxiety disorder (OR 4.2, *p* = 0.000) and alcohol use disorder (OR 5.8, *p* = 0.000) were more likely to report suicidal planning. Individuals with chronic pain were also more likely to have engaged in suicidal planning (OR 4.2, *p* = 0.000).

#### 3.2.3. Suicide Attempts

The lifetime prevalence of a suicide attempt was 1.6%, with a 12-month prevalence of 0.2%. Table 4A depicts the association between suicide attempts and demographic variables, and Table 4B depicts the association between suicide attempts and physical and psychiatric disorders. Suicide attempts were less likely in those aged between 50 and 64 (OR 0.3, *p* = 0.033) and those who were widowed (OR 0.1, *p* = 0.046). Respondents of Indian ethnicity (OR 1.8, *p* = 0.045), those with a highest education level of secondary school (OR 5.2, *p* = 0.002), and those who were divorced/separated (OR 5.2, *p* = 0.002) were more likely to have attempted suicide. This remained significant after controlling for the presence of psychiatric disorders. Those with major depressive disorder (OR 7.0, *p* = 0.000), bipolar disorder (OR 6.6, *p* = 0.000), generalized anxiety disorder (OR 10.1, *p* = 0.000), and alcohol use disorder (OR 4.1, *p* = 0.003) were more likely to have a lifetime history of suicide attempts. Individuals with hypertension (OR 2.9, *p* = 0.010), hyperlipidemia (OR 2.7, *p* = 0.017), diabetes mellitus (OR 3.7, *p* = 0.014), and chronic pain (OR 2.1, *p* = 0.027) were more likely to have attempted suicide, while those with thyroid disease (OR 0.1, *p* = 0.017) were less likely to have attempted suicide.

## 4. Discussion

Suicide is a worldwide concern. A study of suicidality in 17 countries found the lifetime prevalence of suicidal ideation, planning, and attempts to be 9.2%, 3.1%, and 2.7% respectively [15]. Our findings illustrate that Singapore’s suicidality prevalence mirrors what is seen elsewhere.

Our study found that there is an association between components of suicidality and socio-demographic factors, such as being divorced/separated, and age. The extant literature does highlight that individuals who are divorced have a higher risk for suicide, with the risk higher in divorced men than in divorced women, with divorced individuals manifesting higher levels of depression, anxiety, resentment, a loss of self-esteem and eventually a sense that life is not worth living [16]. Our finding of an association of suicidality with younger age is echoed in two other studies [15,17] which illustrated that age was inversely related with each component of suicidality, and that younger age groups were thus at higher risk of suicide, with the prevalence of suicidality reducing into adulthood. There are some explanations for this finding—firstly, individuals from older age groups might have forgotten, or re-interpreted, prior suicidal behaviors [15] and may have under-reported the presence of suicidal ideation, planning or attempts. Secondly, emotional well-being is noted to increase, and become more stable, with advancing age, with this change remaining significant after controlling for physical health and demographic variables—it is postulated that this improvement possibly occurs due to better adaptation and knowledge [18]. These findings were also echoed in another study, which demonstrated a higher level of positive mental health in individuals aged above 40 as compared to those aged 18–29 [19]. This improved sense of emotional well-being and positive mental health may contribute to lower suicidality. Thirdly, compared to older age groups, younger people might lack the capacity to overcome interpersonal crises and may become more despondent because of them, and this might place younger people at a higher risk of suicide [20]. A caveat to our finding is the elevated rate of suicide completion in the elderly [21], which, if taken together with our findings, may be a reflection that elderly individuals may attempt suicide less often but could die from a suicide attempt more frequently.

The lack of association with employment status is surprising, given that unemployment is a well-established risk factor for suicide [7]. One possibility for the lack of an effect may be the support rendered to those in Singapore who are unemployed—social welfare and unemployment protection reduces the strength of association between unemployment and suicide [22]. In addition, the unemployment rate in Singapore is one of the lowest in the world, which could mean that those who are presently unemployed may retain good prospects of securing employment soon. The lack of a consistent association between suicidality and lower household income is contrary to other literature that indicates that financial stress and suicidality are linked [23]. This finding may be linked to other protective factors in our study population.

Our study illustrated that there was a significant association among depression, bipolar disorder, generalized anxiety disorder, and alcohol use disorder, with suicidal ideation, planning, and attempts. These findings are in keeping with what is known in the extant literature [17]. Depression has a strong association with suicidal ideation and attempts, with 48% of a sample of depressed individuals in Australia endorsing suicidal ideation and 16% endorsing suicide attempts [24], with the severity of depressive symptoms correlating significantly with suicidality. These findings were also echoed in SMHS 2010, with 43.6% of individuals with a history of major depressive disorder endorsing suicidal ideation and 12.3% endorsing suicide attempts [25]. This relationship may be explained by hopelessness, which is a symptom that occurs in depressive illness and is strongly associated with suicide [26]. Regarding bipolar disorder, our findings demonstrate higher odds for lifetime suicidal ideation, planning and attempts in individuals with bipolar disorder than in those with depression. These findings are consistent with findings from SMHS 2010, where 31% of respondents with Bipolar Disorder reported suicidal ideation and 11.5% reported past attempts [27]. Taken together with the extant literature, these findings illustrate that suicidal ideation and attempts occur frequently amongst those with bipolar disorder, with suicidality being most pronounced in the depressive phase of illness [28].

Alcohol use may lead to suicide by impairing judgement, increasing impulsivity, and by leading to mood disorders, cognitive deficits, anxiety, and psychotic disorders [29]. Heavy alcohol users have a five-fold higher risk of suicide than those who drink socially [30]. Further analysis to explore the cumulative effects of multiple psychiatric diagnoses on suicidality is worthy of exploration but is beyond the scope of this paper.

Our findings also illustrate a link between suicidality and chronic medical conditions such as diabetes mellitus and chronic pain. Diabetes mellitus may increase suicidality through multiple mechanisms—the accumulation of diabetes related complications and disabilities, the occurrence of adverse events, stress, and easy access to potentially lethal means (e.g., insulin) [31]. Chronic pain was found to have a statistically significant association with suicidal ideation, planning, and attempts. Chronic pain possibly increases suicide risk by increasing levels of hopelessness and a desire to escape from pain via death [32], by heightening the sense of perceived burdensomeness and creating distress in interpersonal relationships [33].

This study has several strengths, including the large sample size of 6126 adult respondents that is representative of the multi-ethnic general population, the use of a validated scale by trained interviewers, which minimized variability between assessors and assessments and the conducting of interviews in a language most familiar to respondents.

There are several limitations to this study. The self-reported nature of suicidality and physical and psychiatric disorders may be affected by recall bias and could lead to under-reporting. The study team addressed this by utilizing cue cards for participants who could read and avoided making direct mentions of potentially sensitive topics. It is also possible that those who would have endorsed suicidal ideation, planning, and attempts avoided participating in SMHS 2016. The sample surveyed was a community-based, household sample and did not include individuals in hospitals, who are expected to have a higher prevalence of psychiatric and physical disorders and suicidality. The effect of overlapping physical and psychiatric disorders on suicidality was also not assessed. In addition, the non-response rate of 30.5% could lead to an underestimation of suicidality. Finally, whilst not a limitation of this study per se, direct comparisons with the findings of SMHS 2010 could not be made as the latter did not administer the suicidality module of the WHO CIDI, and only obtained data on suicidality amongst participants with major depressive disorder. Thus, we are unable to determine whether there are differences in the prevalence of suicidality or its socio-demographic correlates between SMHS 2010 and SHMS 2016.

## 5. Conclusions

The present study demonstrates that 1 in 13 adult Singaporeans have had suicidal ideation at some point in their lives. The study also demonstrates a significant association between suicidality and certain socio-demographic variables, as well as physical disorders and psychiatric disorders. These findings illustrate that certain groups, especially those with disorders such as chronic pain and bipolar disorder, have a higher likelihood for suicidal behavior and need to be actively screened for suicidal ideation and planning. This is especially pertinent presently, given the current COVID-19 pandemic, which is likely to create an increase in socio-economic problems and the prevalence of mental illnesses.

## Figures and Tables

**Table 1 ijerph-18-04365-t001:** Demographic characteristics of the SMHS 2016 Sample.

Sociodemographic Characteristics		N	Unweighted %	Weighted %
**Age Group (years)**	18–34	1707	27.9	30.4
**(Mean = 45.22)**	35–49	1496	24.4	29.6
	50–64	1626	26.5	26.9
	65+	1297	21.2	13.1
**Gender**	Women	3058	49.9	50.4
	Men	3068	50.1	49.6
**Ethnicity**	Chinese	1782	29.1	75.7
	Malay	1990	32.5	12.5
	Indian	1844	30.1	8.7
	Others	510	8.3	3.1
**Marital Status**	Never married	1544	25.2	31.0
	Married	3843	62.7	59.8
	Divorced/Separated	343	5.6	5.2
	Widowed	396	6.5	4.1
**Education**	Primary and Below	1187	19.4	16.3
	Secondary	1648	26.9	23.0
	Pre-U/Junior College	304	5.0	6.0
	Vocational/ITE	508	8.3	6.3
	Diploma	1024	16.7	19.0
	University	1455	23.8	29.4
**Employment**	Employed	4055	66.2	72.0
	Economically inactive *	1716	28.0	22.7
	Unemployed	354	5.8	5.3
**Monthly Household** **Income (SGD)**	Below 2000	1147	21.0	16.5
	2000–3999	1331	24.4	20.0
	4000–5999	1113	20.4	21.4
	6000–9999	1003	18.4	21.8
	10,000 and above	861	15.8	20.3

* Economically Inactive—e.g., students, homemakers, retirees; SGD—Singapore Dollars.

**Table 2 ijerph-18-04365-t002:** (**A**). Socio-demographic factors of suicidal ideation. (**B**). Associations among suicidal ideation, physical, and mental disorders.

(A) Variable	ODDS Ratio	*p*-Value	95% Confidence Interval
**Age Group**			
18–34 (reference)			
35–49	0.785	0.319	0.488–1.264
50–64	0.407	0.001	0.236–0.701
65+	0.278	0.002	0.125–0.618
**Gender**			
Female (reference)			
Male	0.709	0.034	0.517–0.974
**Ethnicity**			
Chinese (reference)			
Malay	0.703	0.028	0.514–0.963
Indian	0.923	0.577	0.696–1.223
Others	1.051	0.826	0.672–1.645
**Education**			
University (reference)			
Primary and Below	1.186	0.633	0.590–2.382
Secondary	1.279	0.330	0.780–2.096
Pre-University/Junior College	1.162	0.681	0.568–2.377
Vocational Institute/ITE	1.243	0.463	0.695–2.225
Diploma	1.162	0.517	0.738–1.830
**Employment**			
Employed (reference)			
Economically Inactive	0.767	0.233	0.497–1.185
Unemployed	1.329	0.306	0.771–2.289
**Marital Status**			
Married (reference)			
Never Married	1.801	0.007	1.176–2.757
Divorced/Separated	3.564	0.000	2.051–6.195
Widowed	1.531	0.362	0.613–3.826
**Household Income (SGD)**			
Below 2000 (reference)			
2000–3999	0.708	0.156	0.440–1.140
4000–5999	0.588	0.042	0.350–0.982
6000–9999	0.775	0.343	0.457–1.312
10,000 and above	0.524	0.037	0.286–0.962
**(B) Variable**	**ODDS Ratio ***	***p*-Value**	**95% Confidence Interval**
**Psychiatric Disorders**			
Without Psychiatric Disorder (reference)			
Major Depressive Disorder	4.793	0.000	3.249–7.069
Bipolar Disorder	10.990	0.000	5.676–21.276
Generalized Anxiety Disorder	10.272	0.000	5.470–19.289
Obsessive Compulsive Disorder	5.376	0.000	3.321–8.702
Alcohol Use Disorder	3.089	0.000	1.875–5.090
**Physical Illness**			
Without Physical Illness (reference)			
Hypertension	1.422	0.131	0.901–2.245
Hyperlipidemia	1.211	0.409	0.769–1.905
Diabetes Mellitus	1.938	0.020	1.112–3.379
Asthma	1.486	0.051	0.998–2.211
Chronic Pain	2.166	0.000	1.555–3.017
Cardiovascular Disease	0.930	0.864	0.409–2.119
Gastrointestinal Ulcer	1.529	0.359	0.617–3.789
Thyroid Disease	2.077	0.034	1.055–4.087
Cancer	2.890	0.020	1.181–7.077

* Odds Ratio was derived using multivariable logistic regression analyses after adjusting for socio-demographic variables.

**Table 3 ijerph-18-04365-t003:** (**A**). Socio-demographic factors of suicidal planning. (**B**). Associations among suicidal planning, physical, and psychiatric disorders.

(A) Variable	Odds Ratio	*p*-Value	95% Confidence Interval
**Age Group**			
18–34 (reference)			
35–49	0.789	0.649	0.284–2.191
50–64	0.522	0.324	0.143–1.902
65+	0.243	0.117	0.041–1.423
**Gender**			
Female (reference)			
Male	0.817	0.548	0.422–1.581
**Ethnicity**			
Chinese (reference)			
Malay	0.690	0.302	0.341–1.396
Indian	1.280	0.403	0.718–2.284
Others	0.756	0.608	0.259–2.206
**Education**			
University (reference)			
Primary and Below	0.954	0.951	0.211–4.320
Secondary	2.280	0.171	0.701–7.415
Pre-University/Junior College	0.232	0.030	0.062–0.869
Vocational Institute/ITE	1.778	0.405	0.459–6.888
Diploma	1.395	0.473	0.561–3.470
**Employment**			
Employed (reference)			
Economically inactive	0.865	0.726	0.383–1.951
Unemployed	1.541	0.440	0.514–4.619
**Marital Status**			
Married (reference)			
Never Married	1.829	0.193	0.737–4.540
Divorced/Separated	3.512	0.036	1.086–11.358
Widowed	0.375	0.290	0.061–2.305
**Household Income (SGD)**			
Below 2000 (reference)			
2000–3999	1.286	0.559	0.554–2.983
4000–5999	0.856	0.767	0.305–2.403
6000–9999	1.814	0.218	0.703–4.680
10,000 and above	0.426	0.174	0.125–1.459
**(B) Variable**	**Odds Ratio ***	***p*** **-Value**	**95% Confidence Interval**
**Psychiatric Disorders**			
Without Psychiatric Disorder (reference)			
Major Depressive Disorder	5.356	0.000	2.513–11.414
Bipolar Disorder	11.016	0.000	4.523–26.832
Generalized Anxiety Disorder	4.436	0.008	1.465–13.428
Obsessive Compulsive Disorder	1.363	0.621	0.399–4.658
Alcohol Use Disorder	5.832	0.000	2.331–13.439
**Physical Illness**			
Without Physical Illness (reference)			
Hypertension	1.951	0.088	0.906–4.201
Hyperlipidemia	0.917	0.862	0.345–2.436
Diabetes Mellitus	2.164	0.185	0.690–6.789
Asthma	1.738	0.143	0.830–3.642
Chronic Pain	4.219	0.000	2.215–8.037
Cardiovascular Disease	1.516	0.647	0.256–8.991
Gastrointestinal Ulcer	2.915	0.243	0.484–17.563
Thyroid Disease	1.544	0.575	0.338–7.054
Cancer	2.736	0.331	0.259–20.827

* Odds ratio was derived using multivariable logistic regression analyses after adjusting for socio-demographic variables.

**Table 4 ijerph-18-04365-t004:** (**A**) Socio-demographic factors of suicidal attempts. (**B**) Associations among suicidal planning, physical, and psychiatric disorders.

(A) Variable	Odds Ratio	*p*-Value	95% Confidence Interval
**Age Group**			
18–34 (reference)			
35–49	0.684	0.408	0.278–1.682
50–64	0.263	0.033	0.077–0.896
65+	0.148	0.058	0.020–1.067
**Gender**			
Female (reference)			
Male	0.693	0.295	0.348–1.378
**Ethnicity**			
Chinese (reference)			
Malay	0.875	0.71	0.432–1.770
Indian	1.824	0.045	1.014–3.280
Others	0.718	0.574	0.225–2.284
**Education**			
University (reference)			
Primary and Below	1.588	0.587	0.299–8.429
Secondary	5.240	0.002	1.824–15.057
Pre-University/Junior College	2.010	0.351	0.442–9.980
Vocational Institute/ITE	1.003	0.996	0.307–3.280
Diploma	1.701	0.307	0.614–4.716
**Employment**			
Employed (reference)			
Economically inactive	0.768	0.542	0.328–1.796
Unemployed	1.104	0.877	0.315–3.867
**Marital Status**			
Married (reference)			
Never Married	1.109	0.809	0.480–2.563
Divorced/Separated	5.187	0.002	1.870–14.387
Widowed	0.110	0.046	0.013–0.959
**Household Income (SGD)**			
Below 2000 (reference)			
2000–3999	0.769	0.593	0.293–2.016
4000–5999	0.654	0.445	0.220–1.947
6000–9999	1.543	0.428	0.528–4.506
10,000 and above	0.599	0.426	0.170–2.111
**(B) Variable**	**Odds Ratio ***	***p*** **-Value**	**95% Confidence Interval**
**Psychiatric Disorders**			
Without Psychiatric Disorder (reference)			
Major Depressive Disorder	6.963	0.000	3.386–14.320
Bipolar Disorder	6.616	0.000	2.428–18.031
Generalized Anxiety Disorder	10.101	0.000	3.863–26.416
Obsessive Compulsive Disorder	2.745	0.053	0.985–7.648
Alcohol Use Disorder	4.087	0.003	1.638–10.198
**Physical Illness**			
Without Physical Illness (reference)			
Hypertension	2.912	0.010	1.291–6.56
Hyperlipidemia	2.674	0.017	1.195–5.984
Diabetes Mellitus	3.745	0.014	1.310–10.708
Asthma	1.426	0.370	0.657–3.098
Chronic Pain	2.130	0.027	1.090–4.164
Cardiovascular Disease	0.894	0.815	0.350–2.283
Gastrointestinal Ulcer	4.230	0.074	0.871–20.551
Thyroid Disease	0.140	0.017	0.028–0.701
Cancer	0.426	0.278	0.091–1.990

* Odds ratio was derived using multivariable logistic regression analyses after adjusting for socio-demographic variables.

## Data Availability

The data that support the findings of this study are available from the corresponding author, Kundadak Ganesh Kudva, upon reasonable request.

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
