# Peer review of "The Relationship between Suicidality and Socio-Demographic Variables, Physical Disorders, and Psychiatric Disorders: Results from the Singapore Mental Health Study 2016"

_ijerph, 2021, doi:10.3390/ijerph18084365_

Round 1

Reviewer 1 Report

Thank you very much for an opportunity to comment on the manuscript reporting on the relationship between suicidality and socio-demographic variables, physical and psychiatric disorders based on the Singapore Mental Health Study 2016 data. The manuscript is well written and presents interesting information. Several points for the authors to consider:

Introduction: can you please add information on the epidemiology of suicidal behaviour in Singapore based on the existing literature?

Measures: can you please quote the actual questions pertaining to suicidality, which were included in the WHO-CIDI?

Discussion/Conclusions: can you please comment on the comparison between the SMHS 2010 and SMHS 2016 data on suicide attempts, planning and ideation? Can you also provide information on existing suicide prevention strategies/initiatives in Singapore? 

Author Response

Dear Ma'am/Sir,

Thank you for reviewing the paper. Your questions are answered in turn.

Introduction: can you please add information on the epidemiology of suicidal behaviour in Singapore based on the existing literature?

- The required information has been added to the manuscript.

Measures: can you please quote the actual questions pertaining to suicidality, which were included in the WHO-CIDI?

- the WHO-CIDI Suicidality Module V21.1.4 has been added as Appendix 1

Discussion/Conclusions: can you please comment on the comparison between the SMHS 2010 and SMHS 2016 data on suicide attempts, planning and ideation? Can you also provide information on existing suicide prevention strategies/initiatives in Singapore?  

  • the relevant information has been added on to the manuscript.

Thank you 

Reviewer 2 Report

It was stated that on page#3, that the outcome variables are each of physical or psychiatric disorder and the predictor variable being each of suicidal ideation, planning and attempts. This is incorrect. The outcome variable is supposed to be each of suicidal ideation.

It is not clear that the median or range of values for suicidal ideation, planning and attempts. Authors must provide this information. Given the lifetime prevalence of these measures, suicidal ideation, planning and attempts, is too small, not sure how reliable these estimates would be.

Portions of this analysis and results have appeared in a previous publication in Epidemiology and Psychiatric Series in 2019, particularly the correlates of mental disorders.

It is also important to discuss the results of the previous survey SMHS 2010. How are the results from SMHS 2016 different or similar to SMHS 2010? What additional knowledge learned from previous study? I do not see the authors have contributed any substantial knowledge to the literature beyond we already know.

It is also important to discuss the overlapping conditions reported by study participants. For examples, participants who reported multiple factors – both psychiatric factors and physical illness or multiple psychiatric disorders or multiple physical illness.

Author Response

Dear Ma'am/Sir,

Thank you for reviewing our manuscript. The questions raised have been answered in turn. 

It was stated that on page#3, that the outcome variables are each of physical or psychiatric disorder and the predictor variable being each of suicidal ideation, planning and attempts. This is incorrect. The outcome variable is supposed to be each of suicidal ideation. 

- We have made the necessary edits to the analysis and to the manuscript

It is not clear that the median or range of values for suicidal ideation, planning and attempts. Authors must provide this information. Given the lifetime prevalence of these measures, suicidal ideation, planning and attempts, is too small, not sure how reliable these estimates would be.

- We did not provide median or range of values because these variables are binary variables which only have two possible values (.e ‘0’ and ’1’).

It is also important to discuss the results of the previous survey SMHS 2010. How are the results from SMHS 2016 different or similar to SMHS 2010? What additional knowledge learned from previous study? I do not see the authors have contributed any substantial knowledge to the literature beyond we already know.

- The relevant information has been added to the manuscript

It is also important to discuss the overlapping conditions reported by study participants. For examples, participants who reported multiple factors – both psychiatric factors and physical illness or multiple psychiatric disorders or multiple physical illness.

- we did not look at the effect of overlapping and/or multiple conditions and this has been added as a limitation of our study

Round 2

Reviewer 2 Report

While the authors have addressed the major concerns to my earlier reviews, the concern remains is that what additional information authors have learned different from the results from the previous survey, however. The authors have added percentage distribution of variables from the previous survey (again, conducted by the same authors) in the discussion section, but never elaborated on the lessons learned from the previous survey. I suggest expanding the discussion incorporating this into the discussion section for broader understanding of the issue.

Author Response

Dear Sir/Ma'am,

Thank you for reviewing the revised manuscript. Unlike SMHS 2016, SMHS 2010 did not administer the WHO CIDI Suicidality Module. In SMHS 2010, information on suicidal behaviors was only obtained from those who were diagnosed with Major Depressive Disorder. As such, there was no comparable suicidality data obtained from SMHS 2010 against which we can compare our current findings. Information on suicidality amongst those with Major Depressive Disorder that was obtained in SMHS 2010 has already been reflected in the edits made to our current manuscript.

Thank You

Dr. Ganesh Kudva